# The RNA binding protein IMP3 facilitates tumor immune escape by downregulating the stress-induced ligands ULPB2 and MICB

Dominik Schmiedel, Julie Tai, Rachel Yamin, Orit Berhani, Yoav Bauman, Ofer Mandelboim*

The Lautenberg Center for General and Tumor Immunology, The BioMedical Research Institute Israel Canada of the Faculty of Medicine, The Hebrew University Hadassah Medical School, Jerusalem, Israel

**Abstract** Expression of the stress-induced ligands MICA, MICB and ULBP 1–6 are up-regulated as a cellular response to DNA damage, excessive proliferation or viral infection; thereby, they enable recognition and annihilation by immune cells that express the powerful activating receptor NKG2D. This receptor is present not exclusively, but primarily on NK cells. Knowledge about the regulatory mechanisms controlling ULBP expression is still vague. In this study, we report a direct interaction of the oncogenic RNA binding protein (RBP) IMP3 with ULBP2 mRNA, leading to ULBP2 transcript destabilization and reduced ULBP2 surface expression in several human cell lines. We also discovered that IMP3 indirectly targets MICB with a mechanism functionally distinct from that of ULBP2. Importantly, IMP3-mediated regulation of stress-ligands leads to impaired NK cell recognition of transformed cells. Our findings shed new light on the regulation of NKG2D ligands and on the mechanism of action of a powerful oncogenic RBP, IMP3.

*For correspondence: oferm@ekmd.huji.ac.il

**Competing interests:** The authors declare that no competing interests exist.

## Introduction

The regulation of protein expression is essential for development, differentiation, maintenance, division and death of cells as well as homeostasis of tissues (*Buszczak et al., 2014*). Regulation of protein biogenesis is controlled on transcriptional, translational and post-translational level (*Day and Tuite, 1998*). Critical players in this complex process of protein biogenesis are RNA binding proteins (RPBs) that control many important aspects such as RNA trafficking, stability and translation rate (*Gerstberger et al., 2014*). The RBPs are also involved in immunological recognition (*Kafasla et al., 2014*).

IMP3 (also known as IGF2BP3) is a member of the family of the insulin-like growth factor 2 mRNA-binding proteins. It binds mRNAs via 6 RNA-binding domains consisting of 2 RNA recognition motif domains (RRM) and 4 K Homology domains (KH) (*Müeller-Pillasch et al., 1997*).

Besides the critical role of IMP3 in fetal development (*Mueller-Pillasch et al., 1999*) and fertility (*Li et al., 2014*; *Hammer et al., 2005*), its importance as an oncogene in several kinds of cancer was determined in the last few years: Expression of IMP3 could be observed in colon carcinoma (*Li et al., 2009*; *Lochhead et al., 2012*; *Kumara et al., 2015*), adenocarcinomas (*Bellezza et al., 2009*; *Lu et al., 2009*; *Gao et al., 2014*), urothelial carcinomas (*Sitnikova et al., 2008*; *Xylinas et al., 2014*), lymphomas (*King et al., 2009*; *Tang et al., 2013*; *Hartmann et al., 2012*), renal cell carcinomas (*Hoffmann et al., 2008*; *Jiang et al., 2006*; *2008*), and many more (*Hammer et al., 2005*; *Chen et al., 2013*; *Walter et al., 2009*; *Köbel et al., 2009*; *Zhou et al.,*

**eLife digest** Tumor cells differ from healthy cells in many aspects. Importantly, tumor cells have the ability to divide and grow much faster than normal cells. To protect ourselves from full-grown cancers, our bodies have developed a surveillance system: when a tumor cell starts to divide without restraint, "stress-induced" proteins start to appear on its surface. These proteins help the immune system recognize abnormal or damaged cells, allowing the immune cells to eliminate the defective cells.

Despite this system of protection, a tumor cell sometimes manages to avoid having stress-induced proteins placed on its surface, allowing it to remain undetected by the immune system. By studying several different types of human cancer cells, Schmiedel et al. found that a protein called IMP3 is present in cancer cells but not in healthy cells. Further investigation revealed that IMP3 prevents the production of some stress-induced proteins and stops them moving to the cell surface.

Schmiedel et al. also show that the presence of the IMP3 protein in cancer cells causes nearby immune cells to become much less active. This suggests that developing drugs that block the activity of IMP3 could help the immune system to fight back and destroy cancer cells.

2014). Additionally, high expression of IMP3 often correlates with poor survival prognosis for patients (*Hoffmann et al., 2008*; *Jiang et al., 2006*; *Köbel et al., 2009*; *Yuan et al., 2009*; *Lin et al., 2013*). So far, the role of IMP3 as regulator of cell proliferation, migration and invadopodia formation was mainly studied due to its ability to bind and stabilize mRNAs coding for insulin like growth factor 2 (IGF2) or CD44 (*Liao et al., 2005*; *Vikesaa et al., 2006*). Practically nothing is known on its immune evasion properties.

NK cells are important for the immune surveillance of transformed cells. They belong to the innate immune system (although they possess some adaptive features as well [*Sun et al., 2011*; *Min-Oo et al., 2013*]) and are able to kill transformed cells, virus-infected cells and to interact with bacteria and fungi (*Wu and Lanier, 2003*; *Lodoen and Lanier, 2006*; *Lanier, 2008*; *Seidel et al., 2012*; *Schmidt et al., 2013*; *Koch et al., 2013*). The killing by NK cells is mediated by several NK -activating receptors, among them is NKG2D (*Seidel et al., 2012*; *Koch et al., 2013*; *Bauer et al., 1999*). The ligands of NKG2D are MHC class I polypeptide-related sequences A and B (MICA, MICB) and the family of unique length 16 (UL16) binding proteins 1 – 6 (ULBP 1–6), collectively known as stress-induced ligands (*Elias and Mandelboim, 2012*). The stress-induced ligands are differentially expressed on the cell surface following stresses like viral infection, heat-shock or genotoxic stress (*Raulet et al., 2013*). Accordingly, these ligands are important for immune surveillance and both cancer cells and viruses often suppress stress ligand surface expression (*Elias and Mandelboim, 2012*; *Salih et al., 2002*; *Fuertes et al., 2008*; *Stern-Ginossar and Mandelboim, 2009*). Although mRNAs for some of these stress ligands are found almost in every cell, these ligands are barely expressed on the surface of healthy cells (*Stern-Ginossar and Mandelboim, 2009*). This suggests that the expression of the NKG2D ligands is also controlled at mRNA level, and indeed, we have previously shown that 10 different cellular miRNAs negatively regulate the expression of MICA and MICB proteins (*Stern-Ginossar et al., 2007*; *2009*; *Nachmani et al., 2009*). We further demonstrated that several RBPs are involved in the regulation of MICB expression (*Nachmani et al., 2014*).

In this study, we show a new mechanism of IMP3 that facilitates immune evasion of cancerous cells by downregulation of the NKG2D ligand ULBP2 in a direct manner and MICB in an indirect manner. Thereby, we give new insights into the complex biological processes that are regulated by this powerful oncogene. Notably, we also discovered the first cellular mechanism acting on ULBP2.

## Results

### RBP affinity purification reveals IMP3 as potential binder of ULBP2-3'UTR

Our group has previously demonstrated that the stress-induced ligand MICB is regulated by numerous RBPs that directly affect its stability and expression (*Nachmani et al., 2014*). In order to identify

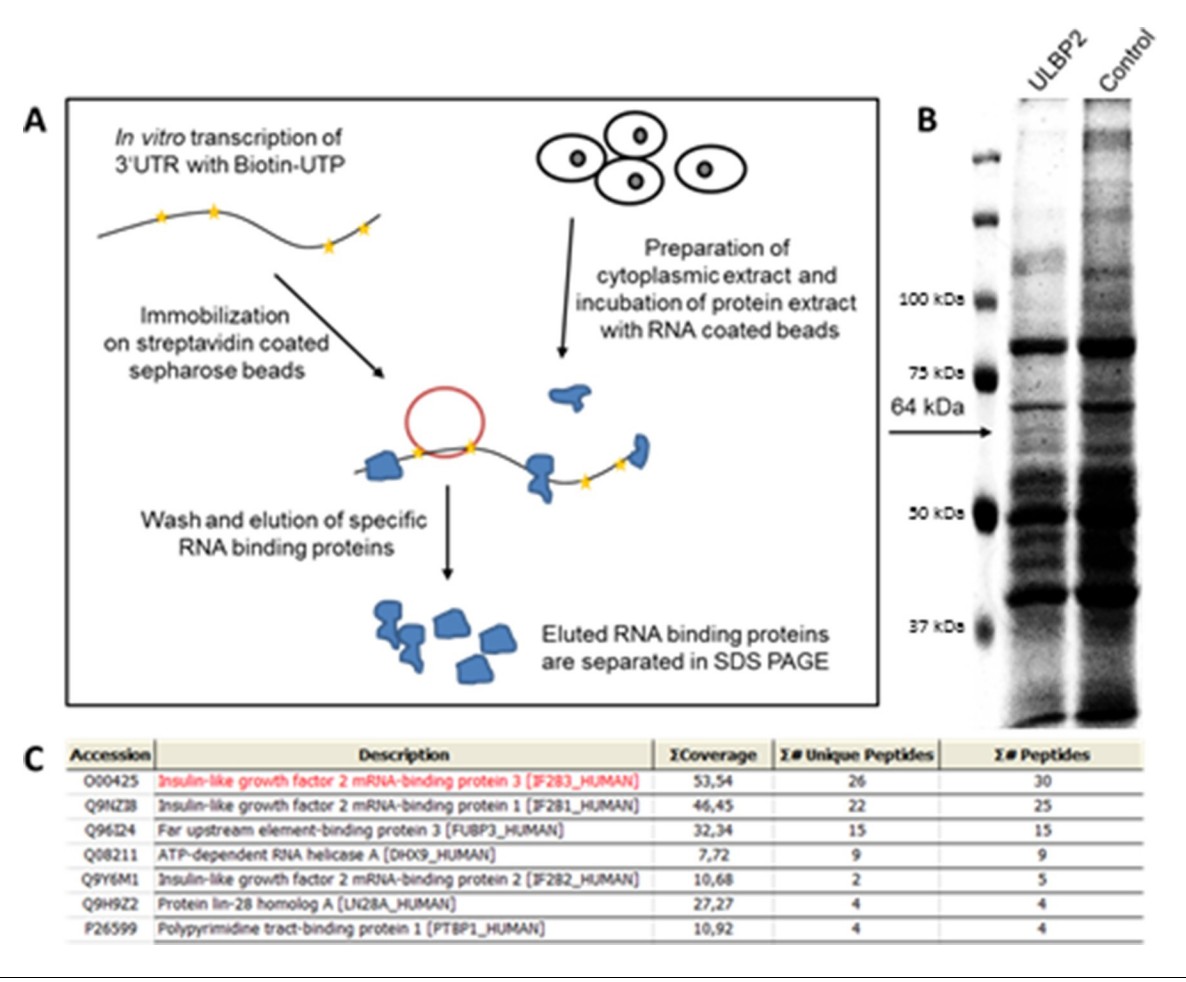

**Figure 1.** RNA affinity purification using the 3 'UTR of ULBP2. (**A**) Schematic representation of the workflow in RNA affinity purification (RNA-AP). (**B**) Cytoplasmic extracts of RKO cells were incubated with RNA coding for the 3'UTR of ULBP2 or control UTRs. The enriched RNA-binding proteins were run on 10% polyacrylamide gel and stained with Coomassie BB G-250. A specific band (indicated with an arrow, 64 kDa) was excised and analyzed by mass spectrometry. For facilitated visualization, a detail of the Coomassie gel is shown in false color and contrast was increased. (**C**) Top-listed results from mass spectrometry analysis of the excised gel part. IMP3 (here called IGF2BP3) is the hit with the highest coverage and amount of specific and total peptides in the analyzed gel band.

additional regulators of other NKG2D ligands, particularly of the ULBP family, we performed RNA binding protein affinity purification (RNA-AP) with subsequent analysis by mass spectrometry (schematically described in *Figure 1A*).

For that purpose, we cloned the 3'UTR sequences of ULBP1, 2 and 3 as well as control UTRs and generated in vitro RNAs in presence of biotin-UTP (*Figure 1A*). About 10 percent of the incorporated UTPs were biotinylated. Next, we generated cytoplasmic extracts of RKO cells (since they express numerous NKG2D ligands, see figures below) and incubated the RNAs with the extracts (*Figure 1A*). We purified the biotinylated RNAs with the specifically bound RBPs using streptavidin-coated sepharose beads and separated the enriched proteins by running the eluates on SDS PAGE gels using reducing conditions. We performed Coomassie staining and excised protein bands that selectively appeared in the lane of ULBP2 (*Figure 1B*, a band at 64 kDa, indicated by an arrow). The band that can be seen in the ULBP2-3'UTR at about 120 kDa was not specific.

Mass spectrometry analysis of an excised gel slice revealed IMP3 (*Figure 1C*, named Insulin-like growth factor mRNA-binding protein 3) as the protein with the highest coverage (percentage of protein sequence covered by identified peptides, named 'ΣCoverage' in *Figure 1C*), the most unique peptides (number of peptides unique to a specific protein, named Σ# Unique peptides in *Figure 1C*)

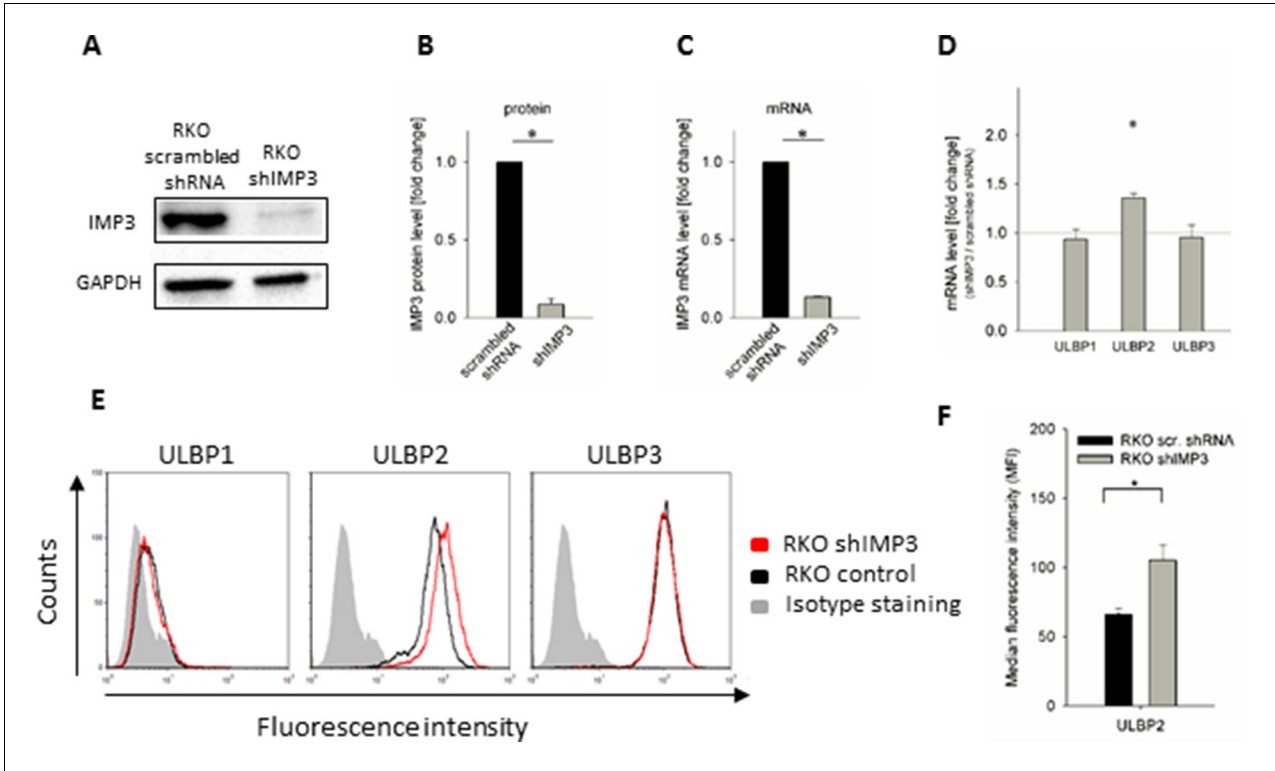

**Figure 2.** Knockdown of IMP3 in RKO cells and surface staining of NKG2D ligands. (**A**) Western Blot analysis of IMP3 (64 kDa) in RKO cells transduced with a scrambled shRNA or with a shRNA against IMP3 (shIMP3). GAPDH (36 kDa) was used as a reference. Sections were cropped. (**B**) Quantification of IMP3 levels of Western Blots assays of two independent experiments performed with two different antibodies recognizing IMP3, relative to GAPDH expression *p=0.02 in student's t-test. (**C**) qRT-PCR analysis of mRNA levels of IMP3 levels in RKO cells transduced with a scrambled shRNA or with shIMP3, normalized to GAPDH. *p<0.0001 in student's t-test. (**D**) qRT-PCR analysis of mRNA levels for the NKG2D ligands in RKO transduced with shIMP3 compared to RKO transduced with scrambled shRNA. Combined data out of 4 replicates, statistics are calculated by comparing transcript levels of the same mRNA in RKO shIMP3 and RKO scrambled shRNA. *p(ULBP2)<0.001 in one-sample t-test. (**E**) Surface expression of NKG2D ligands ULBP1, 2 and 3 on RKO cells analyzed by FACS. Expression is shown on RKO cells transduced with shIMP3 (red histogram) and on cells transduced with a scrambled shRNA (black histogram). The grey filled histogram is the background staining determined for an isotype mouse IgG antibody on shIMP3 RKO. Figure shows one representative experiment out of 3 performed. (**F**) Quantification of ULBP2 surface expression on transduced with a scrambled or a IMP3 targeting shRNA. *p(ULBP2)=0.015

The following figure supplement is available for figure 2:

**Figure supplement 1.** Rescue IMP3 in RKO shIMP3 cells reverses increase in ULBP2 expression.

and total peptides (number of peptides that can be attributed to a particular protein, named Σ# Peptides in *Figure 1C*), suggesting that IMP3 precipitates with the 3'UTR of ULBP2.

## Knockdown of IMP3 increases mRNA and surface expression of ULBP2 in RKO

In order to verify that IMP3 indeed regulates the expression of ULBP2, we transduced RKO cells with a plasmid containing a small hairpin RNA (shRNA) that selectively targets IMP3 (RKO shIMP3). As control, we transduced RKO cells with a scrambled shRNA (RKO scrambled shRNA). We confirmed the knockdown using Western Blot (WB) with two different antibodies targeting IMP3 and used GAPDH as reference (*Figure 2A*, shown is the WB with one of these anti-IMP3 antibodies). Quantification of the WB revealed a remaining protein expression of less than 10 percent (*Figure 2B*). Additionally, we analyzed mRNA levels of IMP3 using qRT-PCR and observed a residual amount of about 13% mRNA of IMP3 compared to scrambled shRNA (*Figure 2C*). Next, we analyzed the mRNA levels of the ULBP ligands in RKO shIMP3 compared to scrambled shRNA and observed a significant increase of ULBP2 mRNA levels following knockdown of IMP3 but no changes in mRNA levels of the

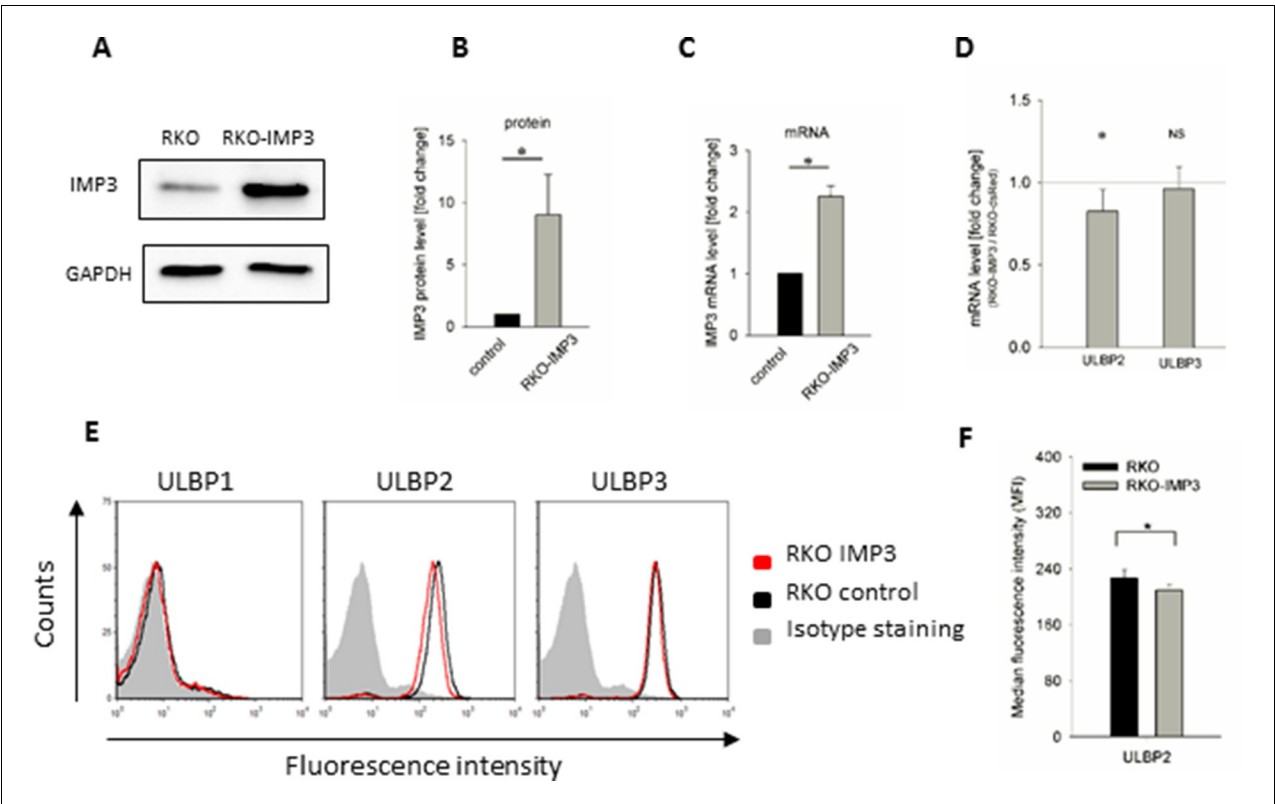

**Figure 3.** Surface staining of stress-induced ligands in IMP3 overexpressing RKO cells. (A) Western Blot analysis of IMP3 in parental RKO cells and RKO cells overexpressing IMP3 (64 kDa). GAPDH (36 kDa) was used as a reference, sections were cropped. (B) Quantification of IMP3 levels of Western Blots assays of two independent experiments performed relative to GAPDH expression. *p<0.03 in student's t-test. (C) IMP3 RNA levels in the RKO-IMP3 compared to RKO analyzed by qRT-PCR. (D) RNA levels of ULBP2 and ULBP3 in RKO-IMP3 compared to RKO-dsRed analyzed by qRT-PCR. *p(ULBP2) =0.044, *p(ULBP3)=0.550 in student's t-test. (E) Surface expression of NKG2D ligands ULBP1, 2 and 3 on RKO cells analyzed by FACS. Expression is shown on RKO cells overexpressing IMP3 (red histogram) and on parental cells (black histogram). The grey filled histogram is the background staining determined for an isotype mouse IgG antibody on parental RKO. Figure shows one representative experiment out of 3 performed. (F) Quantification of ULBP2 surface expression on RKO or RKO-IMP3, *p(ULBP2)=0.034.

other ULBP family members 1 and 3 (*Figure 2D*). To assess the significance of this knockdown, we performed flow cytometry analysis of the NKG2D ligands ULBP1, 2 and 3. We observed an elevation in levels of ULBP2 upon IMP3 knockdown of more than 30%. The surface expression of ULBP3 was unchanged, while little or no expression of ULBP1 was observed on RKO cells prior and after the IMP3 knockdown (*Figure 2E*). We quantified changes in ULBP2 levels in *Figure 2F*. In order to exclude a possible off-target effect of the shRNA that might cause the observed effects on the stress-ligand expression, we performed a rescue experiment of IMP3 expression in RKO (*Figure 2— figure supplement 1*). Expectedly, stress-ligand levels decreased back after the transduction of IMP3 in shIMP3 cells.

## Overexpression of IMP3 causes decrease of surface ULBP2 in RKO

To further validate the findings seen for the knockdown of IMP3 we transduced RKO cells with an IMP3 overexpression vector. We verified the overexpression using WB with GAPDH as reference (*Figure 3A*). The WB quantification revealed a 9-fold overexpression of this RBP (*Figure 3B*). The RNA levels of IMP3 were also significantly increased after transduction (*Figure 3C*). Compared to the knockdown of IMP3, RNA levels and FACS analysis of the overexpression revealed inverse effects on the stress-ligand expression (*Figure 3D and E*). RNA levels in overexpression cells were about 20% reduced compared to the control cells for ULBP2, but not for ULBP3. In FACS analysis, the surface expression of ULBP2 decreased by about 10%. The surface expression of ULBP1 and ULBP3 was unchanged as seen for the knockdown.

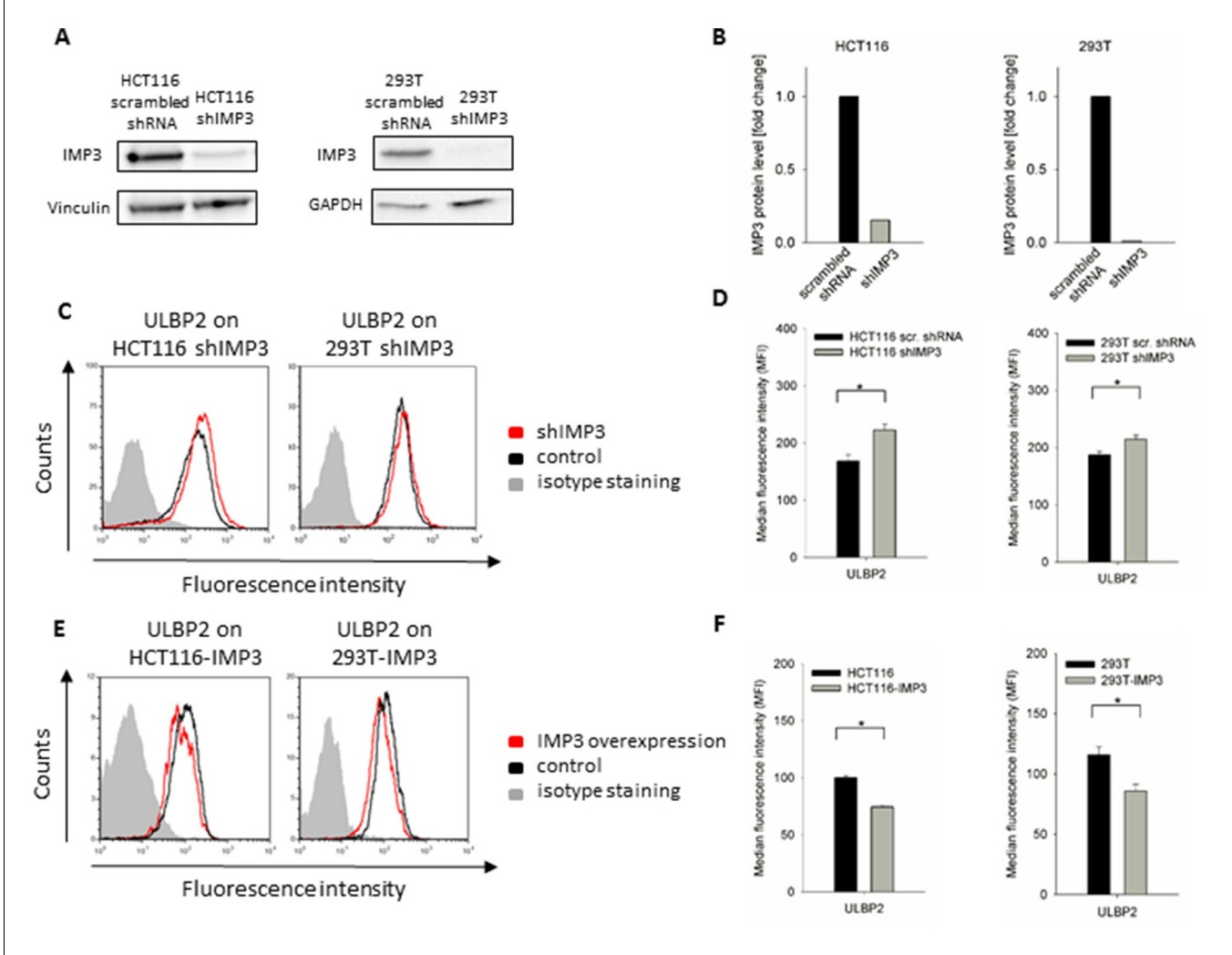

**Figure 4.** ULBP2 after IMP3 knockdown and overexpression in HCT116 and 293T cells. (**A**) Western Blot analysis of IMP3 knockdown in HCT116 and 293T cells compared to cells transduced with a scrambled shRNA. For HCT116 cells, Vinculin was used as reference (130kDa); for 293T cells, GAPDH was used (36kDa). Sections were cropped. (**B**) Quantification of IMP3 WB in HCT116 and 293T, a single experiment was performed. (**C**) FACS analysis of ULBP2 on HCT116 and 293T cell lines with or without IMP3 knockdown. (**D**) Quantification of FACS analysis on HCT116 and 293T cell lines with a knockdown for IMP3 shown if figure C. Statistical analysis for data of three replicates was performed using student's t-test. *p(ULBP2, HCT116)=0.0014, *P(ULBP2, 293T)=8.97 E-4. (**E**) Cell surface staining for ULBP2 in IMP3-overexpressing cell lines HCT116 and 293T. Cells were gated according to their appearance in forward and side scatter and to their GFP levels in FACS that correlate with IMP3 expression. (**F**) Quantification of surface expression of ULBP2 in IMP3 overexpressing HCT116 and 293T cells shown in figure E. Statistical analysis for data of three replicates was performed using student's t-test. *p(ULBP2, HCT116)=9.46 E-9, *p(ULBP2, 293T)=0.0002.

The following figure supplement is available for figure 4:

**Figure supplement 1.** Rescue IMP3 in 293T shIMP3 cells reverses ULBP2 increase.

## Effect of IMP3 on stress ligand expression occurs in several cell lines

To see if the effect of IMP3 on the stress-induced ligands ULBP2 is specific for the colon carcinoma RKO cell line, we performed a knockdown of IMP3 in the colorectal carcinoma HCT116 and in the embryonic kidney-derived cell line 293T and verified the knockdowns using WB. In HCT116 cells, the shRNA targeting IMP3 caused a loss of about 85%; in 293T cells, we could virtually not detect any remaining IMP3 protein (*Figure 4A*, quantified in *Figure 4B*).

Notably, we observed similar effects in all IMP3 knockdown cell lines on stress-ligand expression (*Figure 4C*). In HCT116 colorectal carcinoma cell lines, the knockdown resulted in a significant

increase of ULBP2 of more than 30%; similarly, we observed an increase of about 15% for ULBP2 in 293T upon IMP3 loss this cell line (*Figure 4D*). In order to exclude off-target effects of the shRNA in this cell line as well, we performed a rescue experiment of IMP3 expression in 293T as well (*Figure 4—figure supplement 1*). As we observed for RKO, ULBP2 levels in 293T cells decreased back after the transduction of IMP3 in shIMP3 cells.

To confirm our findings, we performed the overexpression of IMP3 also in HCT116 and 293T cells. In both cell lines, we saw an even stronger decrease of ULBP2 in comparison to RKO after overexpression of IMP3 (*Figure 4D*): 25% for HCT116 and 35% loss for 293T cells. The results are quantified in *Figure 4F*.

## Knockdown of IMP3 increases stability of ULBP2 mRNA

We next investigated the mechanism by which the RBP IMP3 affects ULBP2 expression. To this end, we incubated RKO shIMP3 and RKO scrambled shRNA for 16 hr with the transcription inhibitor D-Actinomycin or with DMSO as diluent control. D-Actinomycin blocks the elongation process of the RNA polymerase II thereby suppressing the generation of new mRNAs. Consequently, a comparison of the RNA levels in the diluent control and the D-Actinomycin treated cells shows the rate of mRNA decay of a specific gene of interest tested in qRT-PCR.

After 16 hr, we isolated total RNA and performed subsequently qRT-PCR analysis on the corresponding cDNAs. We defined the amount of mRNAs in the DMSO-treated control as 1 and calculated the remaining mRNA levels after D-Actinomycin treatment for RKO shIMP3 and RKO scrambled shRNA. Since no new transcripts can be synthesized due to D-Actinomycin treatment, all observed differences are due to alterations in the RNA decay rate. Interestingly, we observed significantly increased amounts of transcript coding for ULBP2 in absence of IMP3 (shIMP3), but – expectedly – no difference in the levels of ULBP3 (*Figure 5*). For ULBP2, about 45% of the initial RNA levels were still present after 16 hr of treatment in the IMP3-knockdown cell lines, but only about

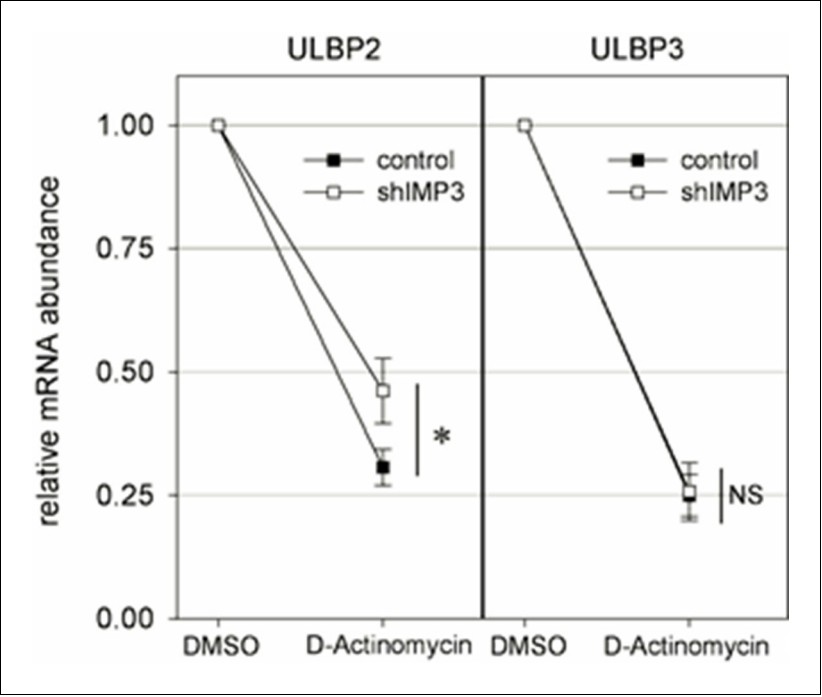

**Figure 5.** Stability determinations of mRNA transcripts of ULBP2 and 3. RKO transfected with a scrambled shRNA (control) or with shIMP3 were treated with D Actinomycin or with DMSO as control. 16 hr later, mRNAs were isolated and cDNA was prepared. The various mRNA transcripts were analyzed using qRT-PCR. Transcript levels were compared by normalization to GAPDH and by setting transcript levels determined for DMSO treatment as 1. Figure shows merged data of three replicates. *p<0.01 for ULBP2 in student's t-test; for ULBP3, no significant differences were observed (NS).

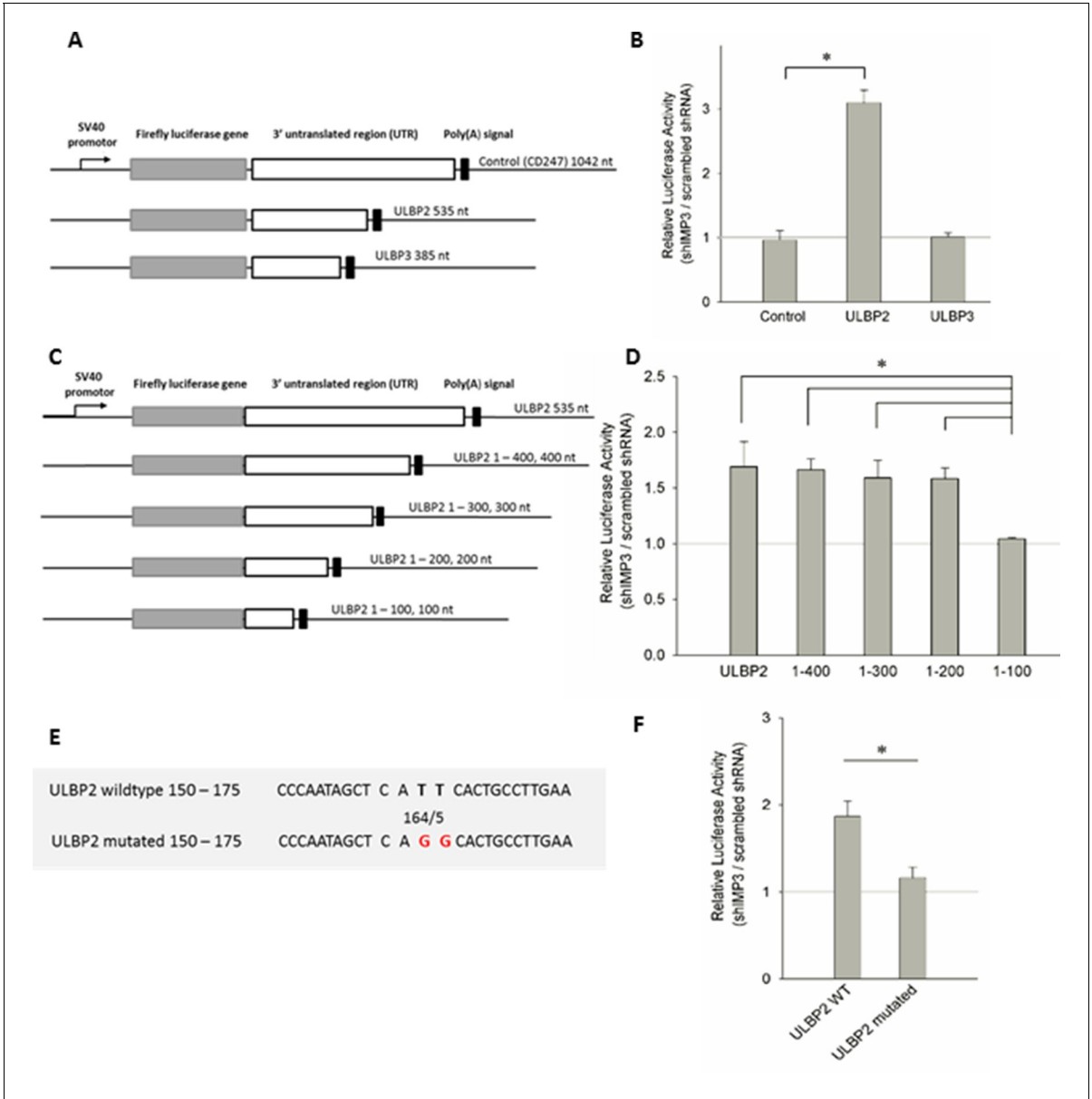

**Figure 6.** Post-transcriptional effects of IMP3 on ULBP2 mRNA assessed by luciferase assay. (A) Schematic representation of the luciferase constructs used for this assay with the 3'UTRs of the ULBP family members including their lengths in nucleotides (nt). (B) The 3'UTRs of ULBP 1, 2, 3 and CD247 (as control) were fused to the luciferase gene as shown in A and expressed in RKO cells transduced either with scrambled shRNA or with shIMP3. 28 hr after transfection of the luciferase vectors, luciferase activity was measured. The results were normalized to empty vector control and statistics were performed based on data acquired for the control UTR (CD247). Figure shows merged data of three independent replicates *p=0.023 in student's t-test. (C) Schematic representation of the truncation mutants of the ULBP2 3'UTR. (D) Luciferase assay with the 3'UTR of ULBP2 and shortened variants presented in (C), expressed in RKO cells transduced either with scrambled shRNA or with shIMP3. Shown is merged data of three independent replicates. The luciferase activity of the fragment ranging from 1–100 bp was significantly lower than the activity of the fragment 1–200 bp (*p=0.002), 1 300 bp (*p=0.006), 1–400 bp (*p=0.041) and the full 3'UTR (*p=0.010) (E) Schematic representation of the ULBP2 3'UTR sequences areas that were mutated. The putative binding motif CATT is shown in loose characters (positions 161 – 164). The introduced mutations in the 3' UTRs (CAGG) are shown in red letters. (F) The wild type (WT) or mutated 3'UTRs (as shown in A) of ULBP2 was transiently expressed in RKO cells transduced with scrambled shRNA or with shRNA IMP3. Luciferase activity was assayed 28 hr after transfection. Shown is merged data of three independent replicates. *p<0.001 in student's t-test.

30% in the RKO cell line with a scrambled shRNA (*Figure 5*). Therefore, we assumed a direct IMP3 binding to ULBP2 mRNA that decreases RNA half-life.

## Identification of the IMP3 binding site in the 3'UTR of ULBP2

To further demonstrate that IMP3 directly interacts with the 3'UTRs of ULBP2, we fused the 3'UTRs of ULBP2 and two control UTRs - ULBP3 and CD247 3'UTRs (we used the 3'UTR of CD247 (CD3ζ) that seemed unlikely to be affected by IMP3) to luciferase genes in the vector pGL3 (schematically described in *Figure 6A*). The constructs were transiently expressed in RKO shIMP3 cells or RKO scrambled shRNA cells and luciferase activity was determined. Following the knockdown of IMP3, we observed a significant increase if the luciferase gene was fused to the ULBP2-3'UTR only (*Figure 6B*). The luciferase activity of all other constructs was not affected. Taking the results of the stability determination in consideration (*Figure 5*) together with the luciferase assay, we concluded that IMP3 decreases directly the mRNA stability of ULBP2.

Next, we wanted to identify the binding site of IMP3 within the 3'UTRs of ULBP2. To narrow down the exact binding site, we decided to perform a continuous shortening of the ULBP2-3'UTR. Out of the full length 3'UTR of 535 base pairs, we generated four short variants covering the range from 1–100, 1–200, 1–300, 1–400 base pairs and compared it to the constructs with the complete 3'UTR (schematically shown in *Figure 6C*). We observed that the construct ranging from 1 to 100 base pairs yielded an equal ratio of luciferase activity in RKO shIMP3 and RKO shRNA scrambled (*Figure 6D*). All other constructs, including the construct ranging from 1 to 200bp, had significantly higher luciferase activities in the IMP3 knockdown cells compared to the shRNA scrambled (*Figure 6D*). Thus, we concluded that the binding site for IMP3 must be located between 100 and 200bp of the 3'UTR of ULBP2.

In 2010, Hafner *et al.* used PAR-CLIP technology to identify putative binding sites of RNA binding proteins and proposed the binding motif 'CAUU' for IMP3 equivalent to CATT on DNA level (*Hafner et al., 2010*). This motif exists twice in the 3'UTR ULBP2, at the positions 161–164 and 292–295 of the 3'UTR. Since we determined that the IMP3 binding site in the 3'UTR of ULBP2 is located between 100 and 200 base pairs (*Figure 6D*), we replaced by PCR the TT nucleotides of the CATT motif found at position 164/165 with GG yielding in CAGG (schematically shown in *Figure 6E*). Consequently, the ULBP2-3'UTR mutation abrogated the effect of IMP3-dependent luciferase activity (*Figure 6F*) completely. Therefore, we concluded from this assay that there is only a single binding site for IMP3 in the 3'UTR of ULBP2.

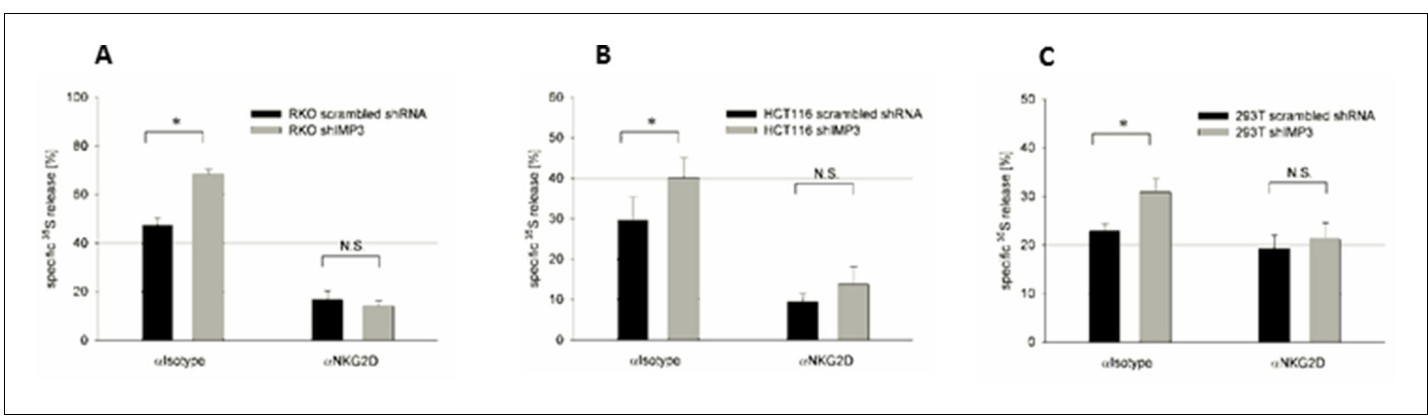

**Figure 7.** Knockdown of IMP3 enhances NK cell-mediated killing of cancer cells in a NKG2D dependent manner. (**A-C**) Primary human NK cells were incubated with an isotype antibody (left columns, αIsotype) or with anti-hNKG2D monoclonal antibody (right column, αNKG2D) for one hour on ice before target cells – either transduced with a control shRNA or shIMP3 – were added. [35]S released into the supernatant upon target cell lysis by NK cells, was assessed 3 hr later (**A**) [35]S release by RKO cells co-cultured with NK cells in the ratio 1:25. *p=0.023 in student's t-test. (**B**) [35]S release by HCT116 cells co-cultured with NK cells in the ratio 1:10. *p=0.001 in student's t-test. (**C**) [35]S release by 293T cells co-cultured with NK cells in the ratio 1:10. *P=0.013 in student's t-test. All experiments were performed at least twice and one representative replicate is shown.

## Cells that express IMP3 evoke a diminished NKG2D-mediated immune response by NK cells

Next, we tested the functional relevance of ULBP2 targeting IMP3. To this end, we co-incubated primary activated bulk NK cells that express the activating receptor NKG2D with RKO, HCT116 and 293T cells expressing shIMP3 or a scrambled shRNA and performed NK cytotoxicity assays. We observed a significantly higher lysis of shIMP3-expressing RKO cells (*Figure 7A*), HCT116 cells (*Figure 7B*) and 293T cells (*Figure 7C*) consistent with the increased surface expression levels of ULBP2 on RKO and HCT116 (*Figure 2E* and *Figure 4B*) and ULBP2 only on 293T (*Figure 4B*). By using a blocking antibody for NKG2D, we demonstrated that the differences observed are due to NKG2D recognition since when NKG2D was blocked killing of the cells was almost identical. The observed drastic decrease in NK cell activation was remarkable taking the moderate shift of ULBP2 following knockdown into account. For that reason, effect of IMP3 on the remaining NKG2D ligands MICA and MICB (MHC class I polypeptide-related sequence A and B) was investigated as well.

## IMP3 affects MICB but not MICA expression in a mechanism different from ULBP2

To assess if IMP3 affects the expression of MICA and MICB, we stained RKO and 293T cells with IMP3 knockdown or a transduced scrambled control for expression of these NKG2D ligands. We found RKO to be negative for MICA but highly positive for MICB. In contrast, 293T cells express MICA but lack MICB expression (*Figure 8A*). Interestingly, we observed an increase of about 50% for MICB following IMP3 knockdown in RKO (quantified in *Figure 8B*), but no effect on MICA. We also validated these results by performing the rescue experiments of IMP3 in these cell lines. In agreement with the KD experiments MICB expression was reduced after the restoration of IMP3 expression in RKO cells and the no effect was seen regarding MICA (*Figure 8—figure supplement 1*). To further confirm that IMP3 affects MICB expression, we overexpressed this RBP in the parental RKO cell line. A dramatic reduction of MICB expression was observed (*Figure 8C*) and only about 20% of the original MICB expression remained (*Figure 8D*). Consistent with our observations for the surface expression of MICB, we could also detect an elevation MICB, but not MICA, RNA levels in RKO cells following IMP3 knockdown (*Figure 8E*). Surprisingly, we could neither detect a IMP3-dependent change in stability of the MICB mRNA using D-Actinomycin treatment (*Figure 8F*) nor IMP3-dependent effects on transcript stability, processing or translation efficacy that would be observed in a luciferase experiment (*Figure 8G*). Consequently, we conclude that the IMP3 uses different mechanisms to affect mRNA and protein levels of ULBP2 and MICB.

## Discussion

The understanding of the diverse mechanistic details of oncogenes in all stages of carcinogenesis: tumor initiation, tumor promotion, malignant conversion and tumor progression (*Multistage Carcinogenesis, 2015*), and the complex interplay of oncogenes with tumor suppressing genes is one of most challenging but also most important objectives in cancer research. Immunotherapy is at the forefront of current cancer research and treatment, with numerous and diverse approaches aimed at harnessing the immune system to fight cancer (*Rosenberg et al., 2004*; *Rosenberg et al., 2008*; *Weiner et al., 2010*; *Mellman et al., 2011*; *Pardoll, 2012*). Accordingly, understanding the regulation of stress-induced ligands has immense importance as they pose potential therapeutic targets on tumor cells.

The stress-induced ligands that are bound by the activating receptor NKG2D seem to play a critical role in immune surveillance and immune escape. Tumors and viruses alike developed numerous mechanisms to avoid surface expression (*Seidel et al., 2012*; *Salih et al., 2002*; *Fuertes et al., 2008*; *Nachmani et al., 2009*; *Fernandez-Messina et al., 2010*; *Nachmani et al., 2010*; *Bauman and Mandelboim, 2011*; *Bauman et al., 2011*) and even to effectively suppress NK cells activity by the release of soluble or exosomal NKG2D ligands (*Fernandez-Messina et al., 2010*; *Kloess et al., 2010*). A complete knowledge of the complex mechanisms that tightly regulate the expression of surface markers in health or completely deregulate the expression in disease is essential to decide about the application of current therapeutic strategies and for the development of new therapeutics which could increase surface expression of these proteins.

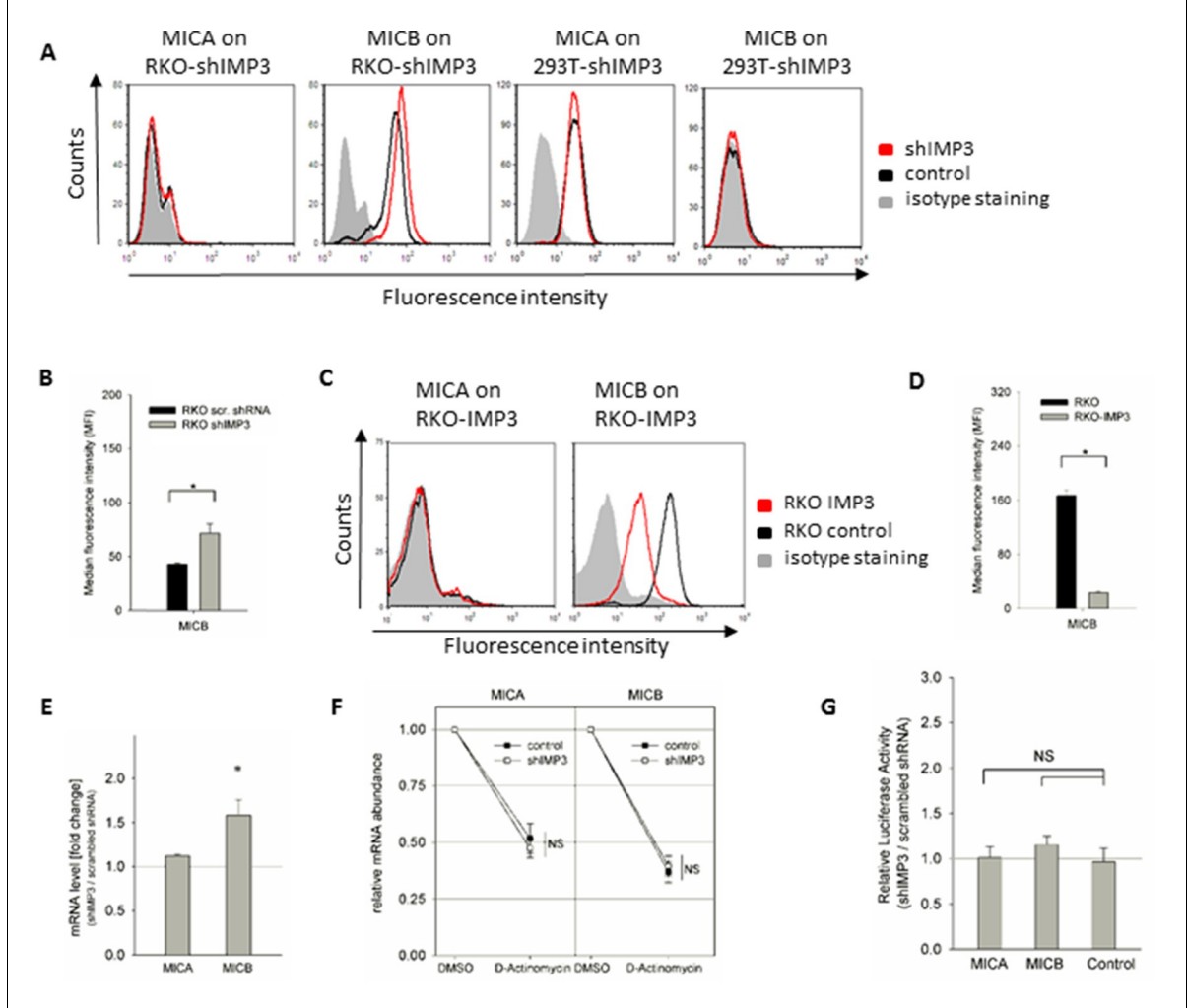

**Figure 8.** IMP3 regulates MICB in a functionally distinct mechanism. (A) FACS analysis of MICA and MICB surface expression on RKO and 293T cells. Expression is shown on cells transduced with shIMP3 (red histogram) and on cells transduced with a scrambled shRNA (black histogram). The grey-filled histogram is the background staining determined for an isotype mouse IgG antibody. Figure shows one representative experiment out of three performed. (B) Quantification of MICB surface expression of RKO cells transduced either with a scrambled or an IMP3 targeting shRNA. *P(MICB)=0.029 in student's t-test (C) FACS analysis of MICB surface protein levels in IMP3 overexpressing RKO cells (red histograms) or controls (black histogram), Shown is one representative experiment of at least three performed ones. (D) Quantification of MICB surface expression of IMP3 overexpressing RKO cells or control cells *P(MICB)=1.24 E-6 in student's t-test. (E) RNA levels of MICA and MICB in RKO transduced with an IMP3 targeting shRNA relative to the scrambled shRNA control and normalized to GAPDH. *P(MICB)=0.0005. (F) RKO transfected with a scrambled shRNA (control) or with an IMP3 targeting shRNA were treated with D-Actinomycin or with DMSO as control. After 16 hr, mRNAs were isolated and cDNA was prepared. The various mRNA transcripts were analyzed using qRT-PCR. Transcript levels were compared by normalization to GAPDH and by setting transcript levels determined for DMSO treatment as 1. Figure shows merged data of three replicates. (G) The 3'UTRs of MICA and MICB and CD247 (as control) were fused to the luciferase gene and expressed in RKO cells transduced either with scrambled shRNA or with shIMP3. 28 hr after transfection of the luciferase vectors, luciferase activity was measured. Results were normalized to empty vector control and statistics were performed based on data acquired for the control UTR (CD247). No significant changes were obtained (NS). Figure shows merged data of three independent replicates.

The following figure supplement is available for figure 8:

**Figure supplement 1.** Rescue IMP3 in RKO and 293T shIMP3 cells reverses MICB downregulation but does not affect MICA expression.

In this study, we present a new mechanism of immune escape that is mediated by the well-established oncogene IMP3. IMP3 is absent in almost all healthy, adult tissues. However, expression of IMP3 could be observed in several kinds of cancer like in colon carcinoma (*Li et al., 2009*; *Lochhead et al., 2012*; *Kumara et al., 2015*), adenocarcinomas (*Bellezza et al., 2009*; *Lu et al.,*

2009; *Gao et al., 2014*), urothelial carcinomas (*Sitnikova et al., 2008*; *Xylinas et al., 2014*), lymphomas (*King et al., 2009*; *Tang et al., 2013*; *Hartmann et al., 2012*) or renal cell carcinomas (*Hoffmann et al., 2008*; *Jiang et al., 2006*; *Jiang et al., 2008*).

We demonstrate in several human cell lines of different origins that the knockdown of IMP3 up-regulates surface expression of the NKG2D ligand ULBP2. Consequently, we observed inverse effects on this stress-ligand following the overexpression of IMP3.

We demonstrate a direct interaction of IMP3 with ULBP2 mRNA. IMP3 was precipitated with biotinylated ULBP2 3' UTR. We observed an increased half-life of ULBP2 mRNA transcripts and, consequently, elevated luciferase activity in absence of IMP3. We also showed that only a single binding site for IMP3 in the 3'UTR of ULBP2 at the position 161–164 exists since its mutation abolished differences in IMP3-dependent luciferase activity.

We tested the effect of IMP3 on ULBP2 in several cell lines using knockdown and overexpression of this oncogene. The effect on ULBP2 expression in RKO, HCT116 and 293T cells were in a comparable magnitude. The impact of IMP3 was capable of altering ULBP2 expression up to 35% in the cell lines tested. However, we also observed an effect of IMP3 on another stress-induced ligand, MICB. Higher expression levels of IMP3 correlated with a reduction of MICB mRNA and surface expression. In contrast to ULBP2, we did not observe differences in half-life or a change in the luciferase activity in presence or absence of IMP3 for MICB. Also, the surface expression changes between knockdown and overexpression for ULBP2 and MICB varied tremendously: In the knockdown, the up-regulation of ULBP2 and MICB was more or less similar, but the IMP3 overexpression decreased only moderately ULBP2 expression, but drastically affected MICB expression. Thus, we concluded that the IMP3-dependant alterations in MICB expression are probably indirect. Most likely, IMP3 acts on a so far unidentified transcription factor or regulator that subsequently leads to alterations in MICB gene transcription. The identification and validation of this putative mediating factor requires further investigation which is beyond the scope of this manuscript.

Finally, we assessed the functional significance in an in vitro system by co-culturing NK cells with RKO cells, HCT116 cells or 293T cells with or without IMP3 knockdown. For all knockdown cell lines, we observed higher killing compared to the cells with natural levels of IMP3.

The physiological significance of IMP3 in healthy, adult tissues is still not completely understood. Obviously, the widely held belief that IMP3 is a strictly oncofetal gene could be disproved by reports that showed expression both in testis and placenta of healthy individuals (*Li et al., 2014*; *Hammer et al., 2005*). The recent discoveries about the role in placental trophoblasts might also shed a light on the array of target genes that are known to be affected by IMP3, for instant CD44 expression that is up-regulated by IMP3 is important in migration of trophoblasts within the placenta. Accordingly, the destabilization of ULBP2 mRNA and regulation of MICB expression described in this study might also protect trophoblasts in the placenta from NK cells, which are the most abundant population of lymphocytes present in the decidua during early pregnancy (*Varla-Leftherioti, 2005*; *Sánchez-Rodríguez et al., 2011*). This assumption requires further experimental support; nevertheless, it would explain the protection of cancerous cells by IMP3 expression that we spotlighted.

Although we could investigate a new effect of IMP3, its role in carcinogenesis is still incompletely understood as well. Our knowledge about its multiple targets that are discovered up to now raise the hypothesis that the complexity of action of an oncogenic RNA-binding protein might exceed the complexity of classical oncogenes like cytosolic or receptor tyrosine kinases. Further research is required to elucidate additional target RNAs that this RNA-binding protein can regulate and to ultimately develop an inhibitor for IMP3 that prevents its impacts on tumor cells.

For cancer therapy, IMP3 might be an excellent drug target due to its restricted occurrence in healthy tissues and its widespread occurrence in different kinds of cancer and especially in cases of unfavorable prognosis. In accordance with our results, an inhibition of IMP3 would - next to other potential benefits - lead to an elevated stress ligand expression and thereby enlighten recent approaches of NKG2D ligand-based cancer treatment, for instance, immunotherapy by Natural Killer cell infusion (*Locatelli et al., 2013*; *Spear et al., 2013*).

# Materials and methods

## Cell lines

All cells were cultivated in 37°C, >95% humidity and 5% $CO_2$ in Dulbecco's Modified Eagle Medium (DMEM, Sigma, Israel) supplemented with 10% heat-inactivated fetal calf serum (Sigma-Aldrich, Israel) and addition of non-essential amino acids, L-glutamine, sodium pyruvate and penicillin/strep-tomycin according to manufacturer's instruction (all Biological Industries, Israel). RKO is a human colon carcinoma cell line (ATCC CRL-2577), HCT116 is a human colorectal carcinoma cell line (ATCC CLL-247), 293T is a human cell line established from embryonic kidney (ATCC CRL-3216). All cells were directly received from the ATCC, Virginia, USA, therefore no further authentication was considered. Prior to transduction, cells were tested negative for mycoplasma contamination. All knock-down and overexpression cell lines showed good viability and proliferation.

## RNA affinity purification and mass spectrometry

The interactions between RNA and RNA-binding proteins were analyzed by RNA affinity purification as previously described (*Hämmerle et al., 2013*). In short, the 3'UTRs of ULBP2 (both sense and antisense) was cloned into pBluescriptII vector. Additionally, a UTR of a control gene with similar length and GC-content (GBP2) was used. In vitro RNA transcription was performed using the MEGA-script T7 transcription kit (Life Technologies, CA, USA) after linearization of the plasmids with PspOMI restriction enzyme (Thermo Scientific (Fermentas), MA, USA). Around 10 percent of totally incorporated UTPs were Biotin-16-UTPs (GE Healthcare, UK). The biotinylated RNAs were coupled with streptavidin-sepharose beads (GE Healthcare, UK) and incubated with cytoplasmic extracts prepared from 80% confluent RKO cells for at least 12 hr. After purification and elution of proteins that bound specifically to the RNAs, a SDS gel analysis was performed and specific bands detected with Coomassie Brilliant Blue G-250 (Sigma Aldrich, Israel). Specific bands were excised and analyzed by mass spectrometry using either a LTQ Orbitrap or a Q Exative LC-MS/MS (Thermo Scientific, Israel). Analysis was performed by the Smoler Proteomics Center, Haifa, Israel.

## Knockdown of IMP3

Knockdown of IMP3 was executed with MISSION shRNA clones (Sigma Aldrich, Israel) for IMP3 in the vector pLKO (MISSION clone: NM_006547.2-2284s21c1 with following sequence: CCGGTGTTG TAGTCTCACAGTATAACTCGAGTTATACTGTGAGACTACAACATTTTTG). IMP3 mRNA is targeted within the 3'UTR (3' untranslated region). Lentiviruses were generated in 293T cells and used for transduction of RKO, 293T and HCT116 that express IMP3 as detected by Western Blot analysis. Next to the transduction of a shRNA specifically targeting IMP3 (shIMP3), a control vector containing a scrambled shRNA was transduced (hairpin sequence: CCTAAGGTTAAGTCGCCCTCGCTCGAGC-GAGGGCGACTTAACCTTAGG). Transduced cells were selected using 2.5 µg/mL puromycin in DMEM for HCT116 and 293T cells and 3 µg/mL for RKO cells.

## Overexpression of IMP3

The vector containing the coding sequence for IMP3 was a kind donation of Prof. Joel K Yisraeli (Hebrew University, Jerusalem, Israel). The insert was transferred into a lentiviral vector and stably transfected into RKO cells. An empty vector served as control. The efficacy of the transduction could be determined using the GFP reporter on the vector. Using limiting dilution, a single RKO clone expressing high GFP was selected and used in all assays described. The overexpression was con-firmed using Western Blot. In order to confirm the effects observed in RKO cells, IMP3 was over-expressed as well in HCT116 and 293T cell line without selection of GFP expressing cells.

## Rescue of IMP3

To rescue IMP3 expression in RKO and 293T, cells that stably express the shRNA targeting IMP3 were transduced with an empty vector as control or the vector containing the coding DNA sequence (CDS) of IMP3. Since the shRNA against IMP3 targets the 3'UTR of the mRNA, the overexpression using the CDS only can't be targeted and downregulated.

## Western blot analysis

Lysates of the various RKO, HCT116 and 293T cells were prepared and SDS gel electrophoresis was executed. Proteins were transferred onto a nitrocellulose membrane with the tank blot procedure and specific protein bands were detected using antibodies detecting IMP3 (sc-47893, Santa Cruz, TX, USA (1:200 in 5% BSA in PBS) and 07–104, Millipore, MA, USA [1:2000 in 5% BSA in PBS]) or GAPDH (Santa Cruz, TX, USA [1:1000 in 5% BSA in PBS]) or Vinculin (Abcam, UK, [1:1000 in 5% BSA in PBS]) as loading control. Chemiluminesce caused by detection antibody-linked horse-reddish peroxidase (HRP, Jackson ImmunoResearch, PA, USA) was detected.

## FACS analysis

For Fluorescent Activated Cell Sorting (FACS) staining of NKG2D ligands on RKO cells, $3 \times 10^5$ were seeded 18 hr prior to analysis in 6-well plates. On the day of analysis, cells were diluted in FACS buffer (PBS, 1% BSA, 0.05% NaN$_3$) and about $1 \times 10^5$ cells were stained with 0.25 μg of the antibody of interest for 1 hr. The cells were always analyzed in a confluence of about 60–80%. Anti-hULBP1, anti-hULBP2, anti-hULBP3, anti-MICA and anti-MICB antibodies were obtained from R&DSystems (MN, USA) as well as the mouse isotype IgG antibody. For detection, a goat-anti-mouse IgG antibody coupled to Alexa Fluor 488 or Alexa Fluor 647 (Jackson ImmunoResearch, PA, USA) was incubated in a dilution of 1:250 for 1 hr. Analysis was performed with a FACSCalibur flow cytometer (BD Biosciences, CA, USA). Cells were gated according to their appearance in forward and side scatter (FSC/SSC); to analyze the effects of the overexpression of IMP3 on stress-ligand expression, only GFP-positive cells were gated (both for the overexpression and the rescue of IMP3 after knockdown in RKO and 293T).

## RNA extraction and cDNA preparation

RNAs for the detection of mRNA levels were prepared using the QuickRNA Kit (Zymo Research, CA, USA). For the generation of cDNA, M-MLV reverse transcriptase (Invitrogen) was used in the presence of anchored Oligo dT primers (Thermo Scientific (Fermentas), MA, USA). Both procedures were performed according to manufacturers' protocols.

## Quantitative Real Time-PCR

For quantitative Real Time-PCR, freshly prepared cDNAs were used for SYBR Green-based detection in a QuantStudio 12k Flex Real-time PCR cycler (Life Technologies, CA, USA) with primers targeting GAPDH, HPRT, ULBP1, ULBP2, ULBP3, IMP3, MICA and MICB.

GAPDH forward: GAGTCAACGGATTTGGTCGTGAPDH reverse: GATCTCGCTCCTGGAAGATG HPRT forward: TGACACTGGCAAAACAATGCA HPRT reverse: GGTCCTTTTCACCAGCAAGCT ULBP1 forward: GCGTTCCTTCTGTGCCTC ULBP1 reverse: GGCCTTGAACTTCACACCAC ULBP2 forward: CCCTGGGGAAGAAACTAAATGTC ULBP2 reverse: ACTGAACTGCCAAGATCCACTGC ULBP3 forward: AGATGCCTGGGGAAAACAACTG ULBP3 reverse: GTATCCATCGGCTTCACAC TCAC IMP3 forward: AGACACCTGATGAGAATGACC IMP3 reverse: GTTTCCTGAGCCTTTACTTCC MICA forward: ATCTTCCCTTTTGCACCTCC MICA reverse: AACCCTGACTGCACAGATCC MICB forward: CTGCTGTTTCTGGCCGTC MICB reverse: ACAGATCCATCCTGGGACAG

## Luciferase assay

The 3'UTRs of MICA, MICB, ULBP2, ULBP3 and CD247 were cloned downstream to a Firefly reporter cassette in the vector pGL3 (Promega, WI, USA) as described (*Stern-Ginossar et al., 2007*). The vector contains a SV40 promoter replacing the natural gene promoter and a SV40 polyadenylation site. $1.25 \times 10^5$ RKO cells transfected with scrambled or IMP3-specific shRNA were seeded into 24-well-plates. The next day, the cells were transfected using TransIT-LT1 reagent (MIRUS Bio, WI, USA) with 250 ng of Firefly-Luciferase vector containing the 3'UTRs, and 50 ng of Renilla Luciferase vector (pRL-CMV) as reference. 28 hr post transfection, cells were lysed and luciferase activity measured using Dual-Luciferase Reporter Assay System (Promega, WI, USA). For the assessment of differences of luciferase activity of the shortened 3'UTRs of ULBP2, following UTRs were cloned into pGL3: 1–100 bp (basepairs), 1–200 bp, 1–300 bp, 1–400 bp and full length UTR (540 bp). $1 \times 10^5$ RKO cells were seeded and the following day transfected with constructs coding for these shortened UTRs. Cells were harvested and analyzed 28 hr post transfection. For the binding site analysis of IMP3 in

ULBP2-3'UTR, the putative-binding motif CATT at position 162 – 165 was mutated into CAGG to prevent binding of IMP3 to the RNA. 1 x $10^5$ RKO cells were transfected with the unchanged, wild type UTR of ULBP2 or the CAGG-mutated UTR and harvested 28 hr post transfection. The empty vector pGL3 was used as control and for normalization. For the readout, a LB 940 Mithras device (Berthold Technologies, Germany) was used. For technical reasons, the device needed to be exchanged between the performance of Figure 6B and Figure 6D/Figure 6F.

### D-Actinomycin treatment

For the transcription inhibition with D-Actinomycin, RKO cells transfected with a scrambled or IMP3 specific shRNA were seeded in a density of 2.5 x $10^5$ in 6-well-plates. The next day, medium was exchanged to fresh DMEM containing either 5 µg/mL D-Actinomycin (Sigma Aldrich, Israel) or equal volume DMSO as diluent control. D-Actinomycin effectively inhibits the elongation process of the RNA polymerase II, thereby suppressing the synthesis of new mRNA transcripts (*Sobell, 1985*). After 16 hr incubation, medium was removed, and cells were harvested following the manufacturers' protocol for RNA extraction (Zymo Research, CA, USA). cDNA was prepared using M-MLV reverse transcriptase (Invitrogen, CA, USA) according to manufacturer's protocol. qRT-PCR was performed as mentioned in the corresponding section. For analysis, we defined the level of a specific mRNA in the DMSO-treated control cells (diluent control) in both the scrambled shRNA- and shIMP3-transduced cells as 1. By calculating the ratio of mRNA level after 16 hr D-Actinomycin treatment and 16 hr DMSO treatment, we assessed the rate of mRNA decay in dependence of IMP3.

### $^{35}$S release assay

NK cells were purified from whole blood of healthy donors via MACS separation (Miltenyi Biotec, Germany). Separated NK cells were activated by co-cultivation with 50 000 irradiated PBMCs of two autologous donors, 5 000 irradiated 8866 B cell lymphoma cells and 0.2 µg/mL phytohemagglutinin (PHA, DYN diagnostics, Israel). NK cells were further cultivated for three weeks in IL-2 containing medium.

For the $^{35}$S release assay, 2 x $10^5$ of the transduced RKO, HCT116 and 293T cells were labeled with 1 µCi/mL EasyTag L-[35S]-Methionine (Perkin Elmer, Israel) in methionine-free RPMI for about 12 hr. After removing excess labeling solution, NK cells were co-incubated with the labeled RKO cells in the ratio 25:1, with labeled HCT116 or 293T cells in the ratio 10:1. Before the assay was set up, NK cells were blocked for 1 hr on ice with 1 µg IgG1 isotype control or a blocking NKG2D antibody. After about 3 hr co-cultivation of cells, plates were centrifuged to pellet cells and 50 µL of the supernatants were transferred to Optiplate-96-plates (Perkin Elmer). 150 µL of Microscint-40 solution (Perkin Elmer) per well were added. The readout was performed using a MicroBeta2 device (Perkin Elmer). The percentage of specifically lysed cells in a certain well was calculated according to following formula: [%] specific lysis = ((count – 'spontaneous') / ('total' – 'spontaneous')) *100%. Spontaneous release was assessed by culturing labeled RKO cells without NK cells; total release was assessed by adding 150 µl 1M NaOH to RKO cells to lyse all cells.

### Statistical analysis

Unless stated otherwise, student's t-tests were applied for statistical analysis. P-values are stated in the corresponding figure legends. The statement 'replicate' defines a biological replicate meaning that cells between experiments were separately seeded and grown and antibodies were diluted discretely. The number of technical replicates within a biological replicate is one for FACS analyses and four for qRT-PCR analyses, luciferase experiments and $^{35}$S release assay.

## Acknowledgements

We thank the Department of Genetics in the Hadassah Medical School in Jerusalem for kind assistance with the luminometer.

# Additional information

## Funding

| Funder | Grant reference number | Author |
|---|---|---|
| Israel Science Foundation | | Ofer Mandelboim |
| European Research Council | | Ofer Mandelboim |
| German-Israeli Foundation for Scientific Research and Development | | Ofer Mandelboim |
| Seventh Framework Programme | FP7-PEOPLE-2012-ITN-317013 | Dominik Schmiedel Ofer Mandelboim |

The funders had no role in study design, data collection and interpretation, or the decision to submit the work for publication.

## Author contributions

DS, Conception and design, Acquisition of data, Analysis and interpretation of data, Drafting or revising the article; JT, RY, OB, YB, Acquisition of data, Analysis and interpretation of data; OM, Conception and design, Analysis and interpretation of data, Drafting or revising the article

## Author ORCIDs

Ofer Mandelboim, http://orcid.org/0000-0002-9354-1855

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
