## [Decision Letter]

Thank you for submitting your work entitled "IMP3 facilitates immune evasion of cancer cells by down-regulating the expression of the NKG2D ligand ULBP2" for consideration by *eLife*. Your article has been favorably evaluated by Sean Morrison (Senior editor) and three reviewers, one of whom is a member of our Board of Reviewing Editors.

The reviewers have discussed the reviews with one another and the Reviewing Editor has drafted this decision to help you prepare a revised submission.

Summary:

Schmiedel and co-workers report on the RNA-binding protein (RBP) IMP3 regulating turnover of ULBP2 transcripts, thereby impacting on ULBP2 surface expression and cytolysis by NK cells. ULBP2 represents one out of eight human stress-induced surface ligands of the activating NKG2D receptor that is expressed on cytotoxic lymphocytes. By mass spectrometry, the authors identified IMP3 as a protein binding in vitro to the 3'UTR of ULBP2 mRNA. A similar unbiased approach has already been used earlier by the authors to describe RPBs for the 3'UTR of another human NKG2D ligand (MICB) (Nachmani et al., 2014; ref. 48). Here, they demonstrate that manipulation of IMP3 expression (shRNA, overexpression) alters cellular expression of ULBP2 in three different cell lines, though the effects appear modest. Functional IMP3 binding to the 3'UTR of ULBP2 transcripts is validated in luciferase reporter assays. Finally, IMP3 down-regulation (by shRNA) is shown to significantly increase susceptibility to NK cytotoxicity in an NKG2D-dependent manner. Overall, the results provide evidence for a novel role of the IMP3 oncogene in cancer immune evasion via down-regulation of ULBP2.

Essential revisions:

The following major concerns were raised that would be required to be adequately addressed for the manuscript to be acceptable for publication in *eLife*.

1) The studies appear to rely on a single shRNA or perhaps a pool of shRNAs (details are not provided), raising concerns of possible off-target effects. Additional experiments are needed to establish that the main findings are observed with more than one independent shRNA and/or that the effects of shRNA knockdown can be rescued by expression of an IMP3 mRNA that does not contain the shRNA targeting sequence.

2) The experiments shown in Figure 7 make the strongest case for a role of IMP3 in regulating cell surface expression of ligands for NKG2D. However, these experiments do not demonstrate that the effect of IMP3 knockdown is due to increased expression of IMP3. In fact, the authors previously showed regulation of MICB by other RNA binding proteins, in Nat Commun. 2014; 5: 4186. This is especially relevant for Figure 7, since the effect of shIMP3 on NK killing is out of proportion to its effect on ULBP2 expression. Therefore, demonstrating specificity of the effect of IMP3 knockdown on ULBP2 would require simultaneous evaluation of the other relevant NKG2D ligands expressed in these cells, e.g., MICA and MICB. Specificity of the effect of shRNAs in reducing IMP3 expression would have to be established as in point 1.

---

## [Author Response]

Essential revisions:

*The following major concerns were raised that would be required to be adequately addressed for the manuscript to be acceptable for publication in eLife. 1) The studies appear to rely on a single shRNA or perhaps a pool of shRNAs (details are not provided), raising concerns of possible off-target effects. Additional experiments are needed to establish that the main findings are observed with more than one independent shRNA and/or that the effects of shRNA knockdown can be rescued by expression of an IMP3 mRNA that does not contain the shRNA targeting sequence.*

As suggested, we overexpressed IMP3 in cells that were stably transfected with a small hairpin RNA (shRNA) targeting IMP3 in order to exclude an off-target effect of the shRNA causing the observed effects on ULBP2 that we describe. Since we transfected a plasmid containing only the coding sequence only for IMP3, the expression of the protein is not affected by the shRNA that targets a part of the mRNA in the untranslated region of the mRNA. Consequently, we observe a downregulation of ULBP2 in the cells with IMP3-rescue as compared to the cells lacking IMP3 proving that the effect is indeed IMP3-specific. The corresponding FACS plot is shown in Figure 2—figure supplement 1 and described in the text. Also, we provided the clone number and hairpin sequence of the shRNA that we used for the knockdown in the Materials and methods section.

2) The experiments shown in Figure 7 make the strongest case for a role of IMP3 in regulating cell surface expression of ligands for NKG2D. However, these experiments do not demonstrate that the effect of IMP3 knockdown is due to increased expression of IMP3. In fact, the authors previously showed regulation of MICB by other RNA binding proteins, in Nat Commun. 2014; 5: 4186. This is especially relevant for Figure 7, since the effect of shIMP3 on NK killing is out of proportion to its effect on ULBP2 expression. Therefore, demonstrating specificity of the effect of IMP3 knockdown on ULBP2 would require simultaneous evaluation of the other relevant NKG2D ligands expressed in these cells, e.g., MICA and MICB. Specificity of the effect of shRNAs in reducing IMP3 expression would have to be established as in point 1.

We thank the editors and reviewers for this comment. We checked the raised question if also MICA and MICB are dependent of the expression of IMP3. Actually, we also observed for MICB a strong IMP3-dependence in surface expression and RNA levels but not for MICA. Thus, as suggested by the reviewers and editors, the NKG2D dependent killing is not only stimulated by increased ULBP2 levels, but also by MICB up-regulation. However, we checked in shIMP3 RKO cells luciferase activity and MICB mRNA stability and concluded from our results that the regulation of ULBP2 and MICB has unequal underlying mechanisms. ULBP2 is directly targeted by IMP3, whereas MICB is not. We added new Figure 8 to the revised manuscript containing all results obtained regarding the expression of MIC family members and discuss our observations amply in the Discussion.